# Qualitative study exploring lessons from Liberia and the UK for building a people-centred resilient health systems response to COVID-19

Rosalind McCollum [1], Zeela Zaizay,[2] Laura Dean,[1] Victoria Watson,[1] Lucy Frith,[3] Yussif Alhassan,[1] Karsor Kollie,[4] Helen Piotrowski,[1] Imelda Bates [5] Rachel Anderson de Cuevas,[6] Rebecca Harris,[6] Shahreen Chowdhury,[1] Hannah Berrian,[7] John Solunta Smith [7] Wede Seekey Tate,[7] Taghreed El Hajj,[5] Kim Ozano,[1] Olivia Hastie,[8] Colleen Parker,[9] Jerry Kollie,[7] Georgina Zawolo,[7] Yan Ding [5] Russell Dacombe,[1] Miriam Taegtmeyer,[1,10] Sally Theobald[1]

RM and ZZ are joint first authors.

For numbered affiliations see end of article.

**Correspondence to**
Dr Rosalind McCollum;
rosalind.mccollum@lstmed.ac.uk

## ABSTRACT

**Introduction** COVID-19 has tested the resilience of health systems globally and exposed existing strengths and weaknesses. We sought to understand health systems COVID-19 adaptations and decision making in Liberia and Merseyside, UK.

**Methods** We used a people-centred approach to carry out qualitative interviews with 24 health decision-makers at national and county level in Liberia and 42 actors at county and hospital level in the UK (Merseyside). We explored health systems' decision-making processes and capacity to adapt and continue essential service delivery in response to COVID-19 in both contexts.

**Results** Study respondents in Liberia and Merseyside had similar experiences in responding to COVID-19, despite significant differences in health systems context, and there is an opportunity for multidirectional learning between the global south and north. The need for early preparedness; strong community engagement; clear communication within the health system and health service delivery adaptations for essential health services emerged strongly in both settings. We found the Foreign, Commonwealth and Development Office (FCDO) principles to have value as a framework for reviewing health systems changes, across settings, in response to a shock such as a pandemic. In addition to the eight original principles, we expanded to include two additional principles: (1) the need for functional structures and mechanisms for preparation and (2) adaptable governance and leadership structures to facilitate timely decision making and response coordination. We find the use of a people-centred approach also has value to prompt policy-makers to consider the acceptance of service adaptations by patients and health workers, and to continue the provision of 'routine services' for individuals during health systems shocks.

**Conclusion** Our study highlights the importance of a people-centred approach, placing the person at the centre of the health system, and value in applying and adapting the FCDO principles across diverse settings.

## STRENGTHS AND LIMITATIONS OF THIS STUDY

⇒ A key strength of this study is the multidirectional learning between health systems in the global south and global north, which involved a wide range of researchers across both settings, and the breadth of perspectives captured from front-line staff and key decision-makers.

⇒ The greatest limitation of this study is that it was carried out at a single point in time, towards the end of the first wave in the UK and before there had been a large increase in cases in Liberia. Response measures have evolved in both settings in subsequent stages of the pandemic.

⇒ The study was limited by the differing range of respondents across study settings, with participants from across a range of health system levels including primary care, hospital front-line workers and decision-makers, as well as regional decision-makers within Merseyside, UK; compared with national and county level decision-makers, technicians and supervisors of front-line staff in Liberia, which may result in differing perspectives.

## INTRODUCTION

The COVID-19 pandemic has forever altered our world. Its impact has been felt across all nations, demonstrating the importance of resilient health systems in protecting global health security.[1] Health systems have been forced to adapt to new ways of working alongside the continued provision of essential services including: prevention of communicable diseases; sexual and reproductive health; care for vulnerable populations; ongoing management of chronic illness (including mental health conditions); continuity of critical inpatient therapies; management of emergency health conditions; and

auxiliary services, including diagnostic imaging, laboratory and transfusion services.[2]

In April 2020, the United Nations expressed concern that, within Africa, up to 3.3 million people could lose their lives as a direct result of COVID-19 and many more through the indirect effects of disruption to health services and worsening socioeconomic conditions.[3] Conditions considered to increase the risk of infection include overcrowded and poorly serviced slum dwellings; limited access to basic handwashing facilities; high levels of informal employment limiting ability to work from home; high levels of malnutrition and lower ratios of beds and health workers to the population.[3] A commentary published by Agyeman *et al* at the outset of the pandemic highlighted a rapid response within many African settings, including a focus on early introduction of screening procedures at ports of entry, and a need for effective community engagement to educate about the mode of transmission. Key protective behaviours were emphasised, along with the need to prepare intensive care beds, and clear government strategies regarding how to deal with hospitalised COVID-19 patients to avoid disrupting the health system and to prevent non-COVID-19-related deaths.[4] Subsequent studies have revealed that indirect health impacts from COVID-19 disproportionately impact women and children.[5 6] Diversion of resources (financial, material, human) from existing health services to address the pandemic, impacts their care.[5 6] This includes supply and demand-side disruptions that can result in lower utilisation of healthcare and, in some cases, impact on quality of care.[7] Bayani *et al* surmise that 'less healthcare will result in more ill health and deaths because health services have been suspended, displaced, or inaccessible.'[7 p.5]

Our study was carried out immediately following the first wave of COVID-19 in Liberia and UK (interviews carried out June to September 2020) in response to an expressed need by stakeholders for this research following dialogue in both contexts. The study was conducted within these two contexts (Merseyside region and Liberia) based on strong prior research relationships within both settings. The differing perspectives from national and county respondents speaking on the national response in Liberia, and front-line health workers and decision makers up to regional level in Merseyside, based on their personal experiences and more localised regional response, is a key limitation. We chose these settings due to the opportunity and demand for research, not because they are exemplars of COVID-19 response. There is, however, still opportunity for learning and comparison on both the strengths and weaknesses within the COVID-19 initial response in both settings. The pandemic has continued to evolve across both settings, with both Liberia and UK experiencing much larger waves of COVID-19 since this original study was carried out. These findings from the first wave can provide valuable lessons to inform continued response to COVID-19 and other health systems shocks.

The pandemic has revealed monopolies of knowledge production, which disempower lower-income and middle-income countries,[8] while pandemic responses in 'developed democracies' have been inadequate, with cuts to health and social services and limited commitment to equity or governance.[8] So-called 'global powerhouses with tried and tested health systems have struggled to contain the COVID-19 pandemic'[9] and health systems have been stretched to the limit, resulting in negative implications for the health of all populations, particularly when access for patients with other acute and chronic illness is limited.[8] As of 1 September 2021, the UK (population 66.8 million)[10] has 6 821 356 confirmed cases and 132 859 COVID-19-related deaths.[11] In the UK, the National Health Service delivers care for most of the population. Meanwhile during the same time period, Liberia (population 4.9 million)[10] has had 5594 confirmed cases, with 245 confirmed COVID-19-related deaths.[11] There are marked differences between settings in the roll-out and scope of testing capacity and uptake of this, with under-reporting in many lower-income and middle-income countries, and so these figures cannot be assumed to be accurate. Future comparisons will eventually show the magnitude of all-cause mortality by age, and firm conclusions can be made about the success of different country approaches. Liberia was initially hailed as one of the top countries in fighting COVID-19, being one of the first countries to start screening at ports of entry (January 2020) and to adopt other control measures such as rapid testing, contact tracing and quarantine.[12 13]

Improving resilience within health systems can build on pre-existing strengths to enhance the readiness of health system actors to respond to crises, while also maintaining core functions.[1] People-centred health systems are a critical framing in shaping resilience as they place people and communities at the centre, while also promoting strategic and collaborative multisectoral leadership which is necessary in delivering a co-ordinated response to a public health crisis.[14] In this paper, we compare health systems responses at a single point in time (June to September 2020) within Monrovia, Liberia and Merseyside, UK, to distil lessons for health systems resilience to a pandemic through comparative case studies which explore aspects of health systems resilience.[15] Within this paper we combine the Foreign, Commonwealth and Development Office (FCDO) eight key principles for promoting resilient health systems with key domains and values of people-centred health systems to frame our findings in relation to the COVID-19 response.[16] Through our discussion we reflect on these expanded principles for resilience against our conceptual framework (figure 1), which is based on a people-centred approach. In response to calls for on-the-ground analysis of the response to COVID-19 within the Global South and comparative case studies that use cocreation and coproduction approaches which go beyond researchers, including policy-makers, practitioners and

**Figure 1** Conceptual framework.

the public,[15 17] we seek to share learning from the response within Liberia and the UK, along with opportunities for multidirectional knowledge sharing.[17] It is our hope that this paper will help inform health policy-makers across global contexts, for the current pandemic response and as they plan towards more resilient people-centred health systems to meet future shocks.

## METHODS

### Study context

Liberia and UK have had very different strategies and case rates from the outset of the pandemic, although there were some similarities in the adoption of infection prevention control (IPC) measures across both contexts. Liberia is among the world's poorest in terms of GDP and living conditions. According to the World Bank 2016 poverty headcount ratio, 44.4% of Liberians live below the international poverty benchmark of US$1.90 per day.[18] The UNDP Human Development Report 2020 ranks Liberia low at 175 out of 189 countries and territories.[19] Inequities between females and males are remarkable with literacy rates (secondary education) of 18.5% and 40.1%, respectively.[19] Liberia has prior experiences of shocks in the form of two civil wars, and the 2014–2015 Ebola virus disease (EVD) epidemic.[20] In response to these experiences, Liberia has prioritised rebuilding a resilient health system, which acknowledges the critical role communities play in addressing their own health needs through the 'Investment Plan for Building a Resilient Health System in Liberia' and the community health services policy (2016–2021).[21 22] By contrast, Merseyside is a Metropolitan County in the North West of England, comprising five boroughs, including the City of Liverpool, including some of the most deprived council areas in England.[23] It has a population of 1.42 million and has had some of the highest numbers of COVID-19 cases in the UK.[24] Within Merseyside, the Liverpool City Region Combined Authority has prioritised tackling deprivation and reducing health inequalities through people-centred care, with integration of health and social care services.[25] Liverpool has a long history of public health innovation, but also a strong sense of local history, culture and place. Throughout the pandemic Liverpool has been at the forefront of community-based innovations and public health strategies, for example, piloting community open access testing for COVID-19.[26]

Liberia introduced stringent border control measures from January 2020, with the establishment of a Special Presidential Advisory Committee on Coronavirus (SPACOC) over 2 months prior to the first recorded cases in the country.[27 28] Liberia's response to COVID-19, prioritised a call to maintain the delivery of routine health services at all levels. Hospitals and clinics continued to provide health services with health facility workers trained in IPC before the first case was identified in country.[28] Physical distancing measures were introduced and use of face masks encouraged.[29]

Within the UK, health service delivery was restructured as part of the COVID-19 response, with routine non-urgent elective care suspended and later restarted in April 2020.[30] Adaptations to minimise potential risk of COVID-19 infection include the use of telemedicine and phone consultations; and changes to essential services for patients, such as changed treatment plans and delays to surgeries.[31] Hospital patient pathways were altered to appropriately triage and cohort the care of COVID-19 patients, reducing the risk of transmission to others and allowing essential services to continue. There was also reduction in routine blood test screening to prioritise COVID-19 PCR testing in response to the UKs 'test and trace' strategy.

### Study aim, design and conceptual framework

Aim: To understand COVID-19 adaptations and decision making in Liberia and Merseyside, UK

This qualitative study explored inductively the differing experiences, perspectives and recommendations of participants in order to understand COVID-19 adaptations and decision-making in Liberia and Merseyside, UK.[32 33] We selected qualitative methods to give 'due emphasis to the meanings, experiences, and views of all the participants'[32] p.43 and understand decision making and the impact of health systems adaptations as a result of COVID-19.

A conceptual framework was jointly developed, following a series of meetings held with researchers in each setting (7 Liberia-based researchers and 18 UK-based researchers). This framework sought to consider a people-centred approach towards the health system's ability to respond to shock, while reflecting the realities experienced in the face of multiple routine challenges (figure 1).[34] The nature of a shock to the health system, whether due to infectious disease outbreak, natural disaster or conflict, influences the rest of the framework.[35] It adopts a people-centred approach at its heart,[14 36 37] while incorporating literature relating to the health system's ability to respond to a sudden shock, and the extent to which it is able to absorb, adapt and transform in response (figure 1).[35 38–42]

People-centred health systems prioritise the collective right to health through integrated and targeted approaches that favour the needs of the most vulnerable.[14 43] Collective action and social solidarity are viewed as essential to the art and science of the development of people-centred systems that are organised around people's healthcare needs and expectations as opposed to diseases, ensuring a continuum of care throughout the life course.[14] This approach embraces the human character of health systems, by viewing individuals, communities and health workers as coproducers of healthcare, placing people and families at the centre.[44] Systems must adapt to meet a range of challenges to support the development of strategies that seek to improve healthcare access and encourage universal coverage. This is particularly important as many individuals transition and oscillate between multiple roles of patient, family and sometimes healthcare provider within one system.

**Table 1** Study participants' role

| Participant role | No of participants interviewed |
|---|---|
| Merseyside, UK | |
| Regional decision-maker | 5 |
| Hospital decision-maker (clinical director, medical director, ward manager) | 4 |
| Hospital consultant | 11 |
| Hospital health worker (junior doctors, nurses) | 10 |
| Health worker in community (GP, district nurse, residential care home) | 7 |
| Liverpool clinical laboratory staff | 5 |
| Total | 42 |
| Liberia participants | |
| National decision-maker | 21 |
| County decision-maker | 3 |
| Total | 24 |

GP, general practitioner.

Interview topic guides were informed by the framework and developed across both settings to explore key areas of health systems functioning in response to COVID-19 (online supplemental appendix 1). Questions included: governance and decision making; use of ethical guidelines; human resource management, infrastructure (information technology and communications) and healthcare worker support; introduction of innovations; and perceptions of the equity and quality of service delivery. Adaptations were made according to the health systems context in each country, for example in Liberia, additional questions were included to explore how learning from the EVD epidemic and other health systems shocks informed COVID-19 response planning.

### Study participants and data collection

The study was carried out at different levels of the health system across both settings (table 1). In Liberia, we conducted key informant interviews in June and July 2020 with 21 national-level and 3 county-level decision-makers (Nimba, Margibi and Montserrado counties) purposively selected because of their involvement with COVID-19 planning and/or routine service delivery. Some had also played key roles in the EVD epidemic response. In Merseyside, we conducted 42 key informant interviews between July and September 2020, with regional, hospital and primary care decision-makers (general practitioners and residential care home managers) and front-line workers selected because of their involvement with COVID-19 planning and/or the delivery of COVID-19 or routine services (see table 1). More interviews were carried out within the UK across health systems levels, due to demand for research across multiple levels and the presence of

a larger team of researchers. In Liberia, by contrast the demand for research was focused at national level, and the research team was smaller in size. The national-level and county-level actors in Liberia, spoke about Liberia's response as a country. In contrast study participants in Merseyside from across health systems levels, including front-line health workers, spoke of their own direct experience within a particular hospital or setting, or on behalf of Merseyside City Region. We acknowledge the limitation that including national-level and county-level actors only within Liberia, creates a somewhat limited perspective. It would have been preferable to have included a larger number and range of participants from subnational health systems levels to provide more depth of understanding about the COVID-19 response.

Interviews were predominantly carried out remotely by researchers experienced in qualitative interviewing in English language, via online platforms such as Microsoft Teams or Skype. A minority were carried out in person with physical distancing measures in place, according to local guidance at the time. All interviews were audiorecorded. Data collection stopped when no new themes emerged from additional data collected.[45] Interviews lasted approximately 30–60 min. Audiorecordings were transcribed verbatim, with quality assurance conducted by a second researcher against the recording.

### Data analysis

The study has sought to use a pragmatic approach to research, working through existing networks to carry out timely research to support the ongoing COVID-19 response in both settings. Both inductive and deductive approaches were blended within data analysis, in keeping with other health systems research.[46–49] In both Liberia and UK, preliminary data analysis workshops were held separately with the research team members involved with data collection. Prior to the workshops all participants reviewed transcripts to familiarise and immerse themselves within the data in order to inductively identify emerging themes which arose from within the study findings. Through these separate country workshops key themes were identified and used to generate a separate coding framework for each setting. All transcripts were imported into NVivo V.12 qualitative data analysis software for coding (QSR International, .12, 2018). Following review of the initial themes which emerged inductively from within the data, there was found to be strong alignment with the eight FCDO principles. These principles were then deductively applied to assist with mapping the findings and enabling comparison between settings. The research team did not simply accept the eight FCDO principles, rather the team reviewed them and found that they did not fully cover all the aspects of resilience which emerged from the data. As a result, two further principles were identified and applied to adequately compare findings between both settings, relating to 'mechanisms for advance preparation' (principle 9) and 'adaptable governance and leadership structures' (principle 10). The

> **Box 1  Expanded principles of health systems resilience in the context of COVID-19 response**
>
> Principle 1 Develop flexible pathways for medical supplies.
> Principle 2 Prioritise a list of essential health services (and continued provision of quality and equitable routine services).
> Principle 3 Build trust with local communities.
> Principle 4 Foster good communication at all system levels.
> Principle 5 Support, recognise and encourage staff.
> Principle 6 Facilitate rapid resource flow and greater flexibility in its use.
> Principle 7 Ensure agile tracking of health information.
> Principle 8 Cultivate effective partnerships and networks.
> Principle 9 Structures and mechanisms for advanced preparedness (New principle).
> Principle 10 Adapt governance and leadership structures to facilitate timely decision-making and effective coordination of response (New principle).

application of the expanded FCDO principles for resilience has helped to showcase how Liberia's experience with responding to prior shocks and their learnt need for early advance preparedness provided an important element working towards resilience. This study is not funded by FCDO, nor were FCDO involved in any way as researchers or coauthors within the research team.

Detailed findings and recommendations were developed into two policy briefs in accordance with these expanded principles for resilience and were shared and discussed with relevant stakeholders from both study settings.[29 50] The relationship of the findings to the original conceptual framework was reviewed and findings compared between settings during a final on-line workshop, attended by all those involved with data collection in both settings, with key similarities and differences jointly discussed.

### Patient and public involvement

Neither patients nor the general public were involved in the design, conduct, reporting or dissemination of our research.

### RESULTS

We present findings according to the expanded FCDO principles for resilience (box 1) (key illustrative quotes are summarised for each principle in table 2). We then reflect on the findings in light of people-centred health systems within the discussion.

Principle 1: Develop flexible pathways for medical supplies: Across both settings supply chains were disturbed due to global shortages and price inflation. In Merseyside there was a lack of personal protective equipment (PPE) and laboratory reagents needed for COVID-19 testing. Meanwhile, in Liberia, the disturbances related to routine supplies as supply chains shifted to focus on COVID-19-related procurement. In both settings, these challenges were felt to relate to global shortages, but were worsened by failure to maintain buffer stocks at local and national levels. In both settings, participants expressed the need for greater decentralisation of procurement decisions.

Principle 2: Prioritise a list of essential health services and continued provision of quality and equitable routine services: Participants from Merseyside expressed fears that there was too much emphasis on COVID-19 care, at times creating redundant capacity, while limiting access and quality of routine essential services. The blanket discontinuation of all elective non-urgent care at the height of the first wave in Merseyside, UK was felt to be unhelpful, and a more nuanced approach which seeks to balance long-term as well as short term risks associated with health conditions was recommended. In contrast, Liberia's early emphasis on routine health services was described as a key learning prioritised by decision-making platforms following the country's experience with the EVD epidemic.

COVID-19 adaptations in the UK led to increased telemedicine, with some respondents raising access-related equity concerns, particularly for elderly populations, who may struggle to engage with telemedicine. There were also concerns raised about quality of care, with some participants in Merseyside fearing delayed-diagnosis, misdiagnosis or suboptimal care due to restrictions limiting physical contact with patients. In Liberia, limited opportunities for supervision, diversion of funds and staff for routine services towards COVID-19 response, and limited community outreach activities (due to physical distancing) were felt to impact quality of care. Across both settings innovations in service delivery have emerged (see policy briefs for details).[29 50]

Principle 3: Build trust with local communities: In both settings, community trust to seek health services declined, which reduced utilisation of services. In Liberia, fear among the population during the start of the pandemic led to reduction in the uptake of health services including national routine vaccination programmes and health facility-based delivery. This was felt to relate to a combination of fear of contracting COVID-19 at facilities and to reduced community outreach activities. Innovative community engagement and social mobilisation strategies were introduced, for example, follow-up visits to pregnant women, which led to patients returning to use services after a few months. Another example is the selective outreach home visits by the neglected tropical disease (NTD) programme to NTD affected patients, in order to avoid interruption in treatment provision. In Merseyside, utilisation of non-COVID-19-related services remained supressed for much longer. This was deemed to relate to widespread community mistrust, and Government campaigns which initially discouraged the public from visiting health facilities via the national 'Stay at home' messaging. Applying learning from Liberia's experience with EVD, the Government of Liberia placed a strong emphasis on working alongside community governance structures, involving local authorities as part of COVID-19 response.

**Table 2** Illustrative quotations from Liberia and Merseyside related to each FCDO Principle

| Principle | Comparison | Quotations |
|---|---|---|
| Principle 1: Develop flexible pathways for medical supplies | Supply chains disturbed across settings due to global shortages and price inflation. Lack of buffer stock in both settings. Restructuring of supply chains in Liberia led to disturbance for routine supplies. | 'Supply chain are affected greatly because their concentration is on how to provide the COVID response activities meaning the …medicines and medical supplies that are needed [for] NTDs (Neglected Tropical Diseases), lack of attention will now be paid to that.' (LIB national decision maker 029)<br>'With regards to PPE, there was national guidance about what we should do and there was a huge amount of fear amongst nurses and medics and everyone else understandably. Everyone was scared. I was scared. If someone said they weren't scared, then they're lying or they're a fool. The national guidance was confused, and availability of PPE fluctuated. Procurement here [NHS hospital] did a very good job, but sometimes it just wasn't delivered nationally. And we went through other supply chains…' (LIV hospital decision maker, Merseyside UK 014) |
| Principle 2: Prioritise a list of essential health services (and continued provision of quality and equitable routine services) | Discontinuation of elective non-urgent care in UK, contrasts with early emphasis on continued routine care in Liberia. | 'So we just have to be robust and do the necessary investment into routine health services, preventive in terms of creating awareness and education among health workers about COVID and how we can continue to care for our patients, with fighting the infection at the same time.' (LIB national decision maker 001)<br>'There's the whole big risk around the screening program…the screening program was stopped, restarting that it's gonna be really challenging. And I suppose that's another risk in terms of people with delayed diagnosis and the right treatment, as a result of not having had that screening mammograms.' (LIV hospital decision maker Merseyside UK 051) |
| Principle 3: Build trust with local communities | Both settings experiences reduced service utilisation due to loss in community trust. Introduction of innovative follow-up visits to patients led to increased service use in Liberia. | 'Some of the useful things that we have been using from Ebola time is, as I said before, to involve the communities …The community aspect is very important because it will help us for the COVID where communities, family members, all of those at the community level are influential group they will be able to comply like we did in the Ebola.' (LIB national decision maker 005)<br>'The elderly population have been shielding because of comorbidities and all that. I think they probably not being as vocal about things that they're concerned about because they're worried about that they will be asked to come in. They fear that that they will catch COVID when they come here.' (LIV hospital health worker Merseyside UK 048) |
| Principle 4: Foster good communication at all system levels | Expansion of virtual communication in both settings. In Merseyside frequently changing guidance from multiple sources created confusion. | 'One of the things that quickly used to come to me is to able to adapt to working with social media technology and all of that, because that's the first thing if you have to communicate with people in this manner you need to understand zooming, skyping, how to take notes…' (LIB national decision maker 029)<br>'And there's so many different sources of information that say different things from what people hear within the hospital talking to friends on the corridor, that you've got to come out with a consistent message. And I think it took longer than was ideal to get a central source of information…But people need to be told what the situation is rather than try to be falsely reassured sometimes as well.' (LIV hospital decision maker, Merseyside UK 004) |
| Principle 5: Support, recognise and encourage staff | Health worker redeployment was common across settings. Health worker training varied in UK according to cadre. | 'Like take for example, when COVID came some of our workers from the (name) Hospital was recruited to go at the front line and (hospital name) is for routine services so taking employees from there to go at the front line that tells you it kind of understaff… So routine services kind of slow down and every attention was placed on COVID but going forward, with the system in place, routine services have gotten back on its feet.' (LIB national decision maker 010)<br>'And it felt like there was unequal share of knowledge and also an unequal kind of confidence in protective clothing. … And I think the people that spent the most time with the patient, the patient areas, for instance, the healthcare assistants and the cleaning staff didn't have all of the information [at the] beginning or any PPE training.' (LIV hospital health worker Merseyside UK 017) |
| Principle 6: Facilitate rapid resource flow and greater flexibility in it's use | Prior under-investment in health was common across settings. In Merseyside there was increased funding available and removal of bottlenecks, which enabled swifter action. | 'The first thing is, we need ownership by government, ownership is not depending on other countries to provide us the resources, to provide the technical capacity. So that is the best recommendation I would say. The ownership has to be there, resources have to be available and the infrastructure has to be available in terms of being resilient.' (LIB national decision maker 029)<br>'To be honest, it was a fairly novel experience because it was a situation where if we asked we more or less got [funding].' (LIV hospital decision maker, Merseyside UK 004) |

Continued

**Table 2** Continued

| Principle | Comparison | Quotations |
|---|---|---|
| Principle 7: Ensure agile tracking of health information | Data quality reduced in Liberia. In Merseyside increased data was collected, but inadequate data analysis measures were put in place. | 'Another recommendation is that we could include COVID to our regular disease surveillance. Like we have the measles, the Lassa, and thing. I think we should include COVID because COVID maybe all around. Like we included Ebola, there should be a document on COVID that will form part of our regular surveillance.' (LIB county decision maker 024)<br>"…there's some value in looking at the things that we were looking at before COVID, because at least we have some longitudinal data on that so that we can see what the effect of COVID is.' (LIV hospital health worker, Merseyside UK 020) |
| Principle 8: Cultivate effective partnerships and networks | Liberia was able to call on prior decision-making structures (established during Ebola response) to enable swift decisions. Need for stronger engagement between primary and secondary care in Merseyside. | 'Involvement of multi-sectorial stakeholders in the response; that was one major thing that we learned from Ebola. And that has been brought to be on this response, so there has been a spark from the level of the presidency where they have key ministries and agency heads heading pillars on the COVID response, involving the community people.' (LIB national decision maker 028)<br>'I think one thing, it's really highlighted is the divide between hospital and primary care. We didn't work together very well before the epidemic, and we are still not working together very well. And I think if things were to get better, the whole health system needs to work better.' (LIV community-level health worker, Merseyside UK 033) |
| Principle 9: Structures and mechanisms for advanced preparedness | Learning from Ebola prompted rapid preparedness in Liberia, in contrast to Merseyside. | 'If you don't prepare well and you are caught unaware you will have a lot of issues, so we didn't wait for COVID to enter Liberia before we prepositioned basic PPE and those are all part of the preparedness phase.' (LIB county decision maker 026)<br>"It was blatantly obvious that anything we've ever planned for in relation to a pandemic or anything along those lines was not the plans that we needed… So I think going forward there needs to be almost a better planning system in place…it's not just a matter of just saying any pandemic it's about what kind of pandemic.' (LIV hospital decision maker, Merseyside UK 069) |
| Principle 10: Adapt governance and leadership structures to facilitate timely decision making and effective coordination of response | Need for rapid guidance from national level to enable subnational decision making was common in both settings. | 'So, at this point in time we think if you give the resources, put the money in the hands of the county health team to buy what they need, that will be more effective … So, we want decision should be given back to the people on the frontline so that they make the decision rather than a centralized point in Monrovia where people sit and decide for people in the lower level and the people choices made the right kind of thing they might need at that level.' (LIB national decision maker 028)<br>'… we were having to work, to a large extent, in the dark. The amount of guidance that came through nationally and even regionally, was actually relatively limited at that stage and we were having to do what felt like quite a lot of planning in isolation.' (LIV decision maker Merseyside UK 008) |

FCDO, Foreign, Commonwealth and Development Office.

Principle 4: Foster good communication at all system levels: The need for effective communication within the health system appeared to be a significant theme, particularly within findings from Merseyside. The rapidly changing context during the early months of the pandemic created a wealth of daily new information. Virtual forms of communication rapidly expanded in both settings, with WhatsApp and online meeting platforms used extensively. Within Merseyside, referred to challenges such as multiple sources of guidance and communication channels struggling to keep pace with the changing guidance, which at times created contradictory messaging and confusion among health workers. By contrast, Liberia developed a centralised messaging procedure with approval needed from the department of Health Promotion before dissemination. In Merseyside, use of emails were typically less popular with staff as these could often be too long and wordy. Participants expressed limited scope for front-line staff to feedback on the information that had been shared.

Principle 5: Support, recognise and encourage staff: Staff redeployment was common across both settings, contributing to varied workloads. In Liberia, health worker redeployment to COVID-19 treatment centres, alongside largely unchanged utilisation rates contributed to increased workload for remaining health workers responsible for provision of routine services. By contrast in Merseyside, redeployment resulted in over-staffing in certain COVID-19 wards. Although there was disparity between health workers, with nurses experiencing increased workload. Due to the reduced volume of patients seeking routine care in the UK, workload was variable for those providing these services. The degree to which health workers received training about COVID-19 prior to having to manage COVID-19 patients varied between settings, with Liberia carrying out training in identification, isolation and infection, prevention and control, before the first case of COVID-19 arrived in country, as a result of lessons learnt following experiences responding to EVD. By contrast in Merseyside, the roll out

of training varied widely by cadre, with some participants identifying that healthcare assistants and cleaning staff did not receive PPE training until later in the pandemic, compared with doctors and nurses (see table 2).

Anticipated mental health implications for health workers emerged from the Merseyside data, due to high rates of COVID-19 infection, exhaustion and high future anticipated post-traumatic stress disorder. This was associated with fear of making treatment mistakes, stress surrounding patient escalation decision making, anxiety over potential COVID-19 infection (both personal and for family), trauma surrounding high COVID-19 infections and deaths and reduced psychosocial support due to remote working. Measures to support staff well-being were introduced (including counselling, reflective therapy, peer support and mentoring, information made available about local support services), with varied levels of uptake. This was not widely discussed in Liberia. Although measures in Liberia to support staff well-being include psychosocial teams, roaming mental health counsellors providing services to health workers are in place. In Merseyside, community support, strong solidarity and teamwork were considered enablers of staff resilience.

Principle 6: Facilitate rapid resource flow and greater flexibility in its use: Historic underfunding of the health system in both settings has been highlighted by the pandemic. In Merseyside, this was considered to be due to nearly a decade of austerity, which has created weariness and uncertainty; whereas in Liberia it related to perception of reliance on external donors which predated the pandemic. Our findings confirmed the need for adequate funding to ensure the building blocks of the health system have received investment prior to the onset of any shock. With the arrival of the pandemic the availability and flexibility of funding differed between settings. In Merseyside, UK, there was increased central government funding, which was mostly freed of usual bureaucratic checks. Managers noted that the removal of these bottlenecks allowed for swift action and rapid adoption of innovations. Front-line managers' ability to make operational decisions was viewed as central to resilience. In Liberia, however, there was an identified need for greater Government of Liberia ownership. Some sectors of the health system, particularly those which are donor reliant struggled in response to reduced partner support following the pandemic. Initially, funding was not made available, however funds for routine service delivery were reallocated to COVID-19 response, with implications for quality (see principle 2). Participants complained about excessive bureaucracy associated with use of funds, which created delays.

Principle 7: Ensure agile tracking of health information: Health information systems (HIS) were rapidly developed in the UK to collect huge quantities of surveillance data on COVID-19 and essential services. However, there was need for improved skills to usefully interpret this data. Respondents in Liberia stated that regular and timely submission of data, particularly from the community

level had declined since the onset of COVID-19. This was considered to relate to reduced data validation, with decreased supervision visits due to physical distancing. In Merseyside, complex new systems were designed to collect pandemic surveillance data, however, data were frequently not analysed or made readily accessible to staff to influence timely monitoring and quality improvement in services. In Merseyside, respondents also noted that a number of new initiatives were introduced during the pandemic, such as virtual consultations, but have not yet been systematically evaluated.

Principle 8: Cultivate effective partnerships and networks: The need for well-established partnerships emerged in both settings, with Liberia already having clear multisectoral participation in decision-making following the Incident Management System (IMS) developed following EVD. Merseyside data highlighted pre-existing weaknesses in collaboration between primary and secondary/tertiary care have been exacerbated. In both settings, the need for greater engagement with the private sector was affirmed, with respondents from UK highlighting the need for stronger links regarding PPE supply chain shortages and in Liberia the need to strengthen collaboration given perceived weakness in private facility IPC standards. Partnerships were established within Merseyside, in a range of aspects of service delivery, including: regional network of laboratory providers to address equipment challenges and ensure COVID-19 testing; between GPs to create service hubs; between disciplines and departments within hospital to address staff shortages and share information. In Liberia, a reduction in the number of partners providing response support was noted. This was a marked contrast to the EVD response.

Principle 9: Structures and mechanisms for advanced preparedness (Newly identified principle from our findings): Within Liberia in particular, but also in Merseyside, there was discussion about advanced preparedness. Respondents in Liberia emphasised how their experiences with previous shocks, particularly EVD, had facilitated learning around early recognition of the need for preparedness. For instance, there was consensus among respondents that waiting for COVID-19 to reach Liberia before responding would be too late. There was early rapid mobilisation of existing emergency response systems which had been established during the EVD response, including; health check controls and quarantines at border points from January 2020; health worker COVID-19 training before the first confirmed case; enhanced hygiene practices; restriction of physical contact and sustained use of PPE, building on institutional memory gained through the EVD epidemic. In contrast, respondents in Merseyside expressed that the COVID-19 response was impeded by a lack of pandemic preparedness for new emerging infectious diseases.

Principle 10: Adapt governance and leadership structures to facilitate timely decision-making and effective coordination of response (Newly identified principle

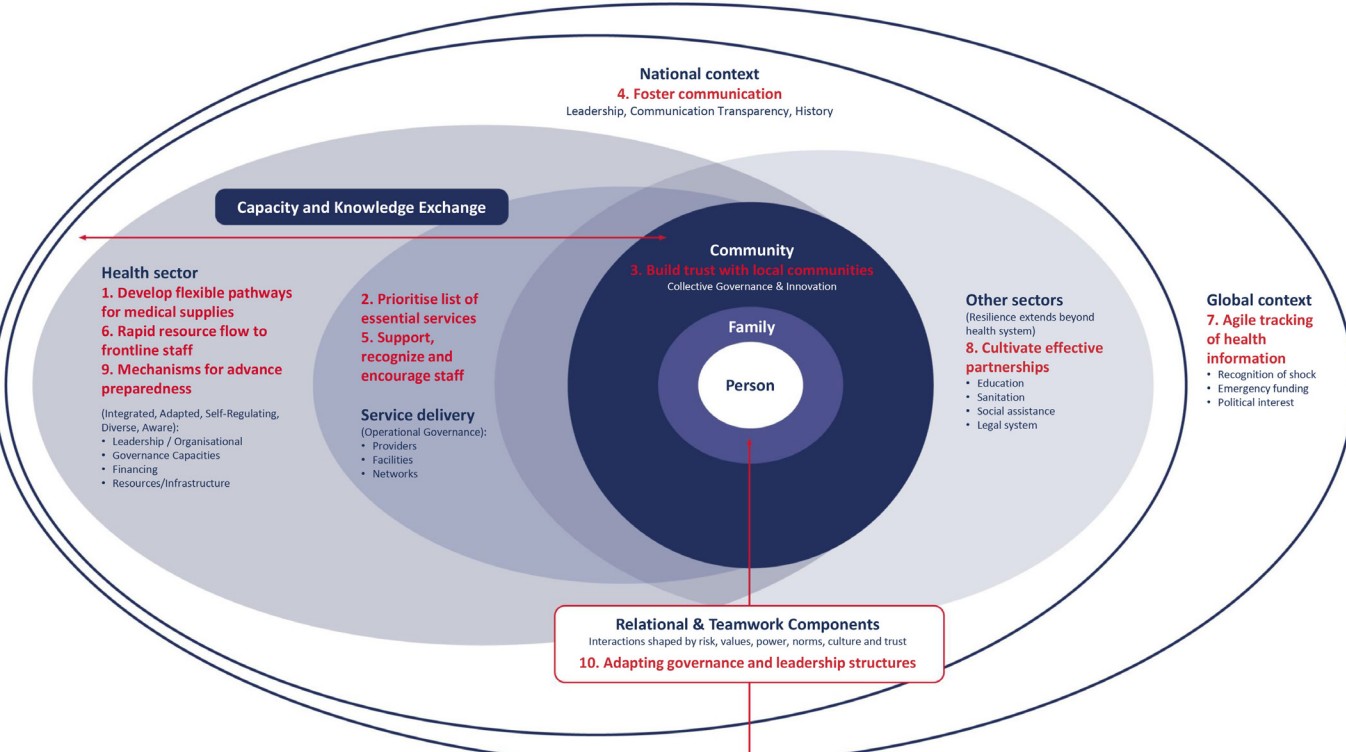

**Figure 2** Expanded principles for resilience and people-centred health systems framework.

from our findings): Being able to adapt governance and leadership structures to facilitate timely response coordination emerged from both settings. Liberia had previously established the IMS in 2014 as part of the response to EVD. It was reactivated in March 2020 to guide planning their pandemic response, led by the Minister of Health. This multisectoral team included a range of political and public health decision-makers, donors and partner representatives. At the time the study was carried out, most decisions were made centrally, with implementation at county level. In Merseyside, early response was hindered by slow and centralised guidance and decision-making, which was perceived to be oriented towards achieving political goals, rather than providing much-needed clarity and recognition of local reality. The limited scope for local autonomy was considered to strain relationships between local senior leadership who sought to enforce central directives and front-line staff, who wanted scope to influence them. In both settings, there was interest in greater decentralisation of decision making to lower levels.

## DISCUSSION

Our findings demonstrate the commonalities between the principles for resilience and people-centred health systems (figure 2). We believe that maintaining a people-centred approach can help ensure that COVID-19 related adaptations are acceptable, understood and meet the needs of individuals (both patients and health workers). The values which underpin people-centred health systems emphasise the need for equity, orienting health services towards a health system which puts 'people and communities at their centre, and surrounds them with responsive services that are coordinated both within and beyond the health sector, irrespectively of country setting and development status.'[14] p. 9

### Adapting a people-centred framework

All ten FCDO principles (eight original principles and two principles identified through this study) are mapped against the original conceptual framework, to demonstrate the connection between our findings and existing literature about resilience (figure 2) and recommendations in response to each principle are outlined in box 2.

### Capacity and knowledge exchange

The continuation of routine essential service delivery following a shock to the health system has previously been highlighted as an area of concern across a range of sectors.[51 52] Health systems need the capacity to continue to deliver services of good quality alongside responding to wider health challenges.[42] Our findings for principle 2 highlighted that COVID-19 adaptations in the UK led to the cancelling or postponing of many essential services, including those related to cancer care, which has been anticipated to decrease life expectancy and survival.[52 53] Meanwhile, Liberia emphasised the need for continuation of routine services and the promotion of patient confidence to use these services. This is in contrast to the EVD epidemic, where over 80% reductions in maternal delivery care in EVD affected areas were described and

**Box 2   Recommendations from expanded Foreign, Commonwealth and Development Office principles for resilience**

1. Supply chains should preposition adequate stocks, diversify sources and seek decentralisation of procurement. Collaboration between providers can prove valuable in securing continuity of supplies.
2. Routine services should be prioritised with a view to long term as well as short-term impact, with prioritisation re-evaluated regularly as the pandemic progresses.
3. Maintain consistent communication and engagement with community leaders, as partners, to participate in pandemic planning within their respective communities.
4. Keep communication channels open, with regular updates for staff which highlight the key information, preferably through meetings, rather than email.
5. Ensure adequate provision of training, with sufficient PPE for health staff, particularly for those staff at highest risk of COVID-19 infection, alongside measures to balance workload and promote staff well-being. Prioritise compassionate leadership which is supportive of staffing levels and rotas, along with staff mental well-being. Investment in psychosocial well-being throughout and after the pandemic response.
6. Health systems need to be adequately funded during 'normal times' if they are to be able to respond when a shock arises. There is urgent need for investment to clear the backlog of delayed routine services.
7. Health information systems need greater investment in both the systems and the human element to be able to analyse, interpret and respond to emerging data trends.
8. Opportunities for multisectoral collaboration should be sought out, with engagement with private sector where possible.
9. Develop a proactive approach, with advance plans for health shocks, along with escalation and de-escalation plans throughout the crisis.
10. Promote greater opportunities for de-centralised staff involvement in decision-making, where feasible. Governments to prioritise an outward focus towards global solidarity.

form part of the reason why routine care was prioritised so strongly as part of the COVID-19 response.[54]

Our findings relating to supply chain (principle 1) resonate with literature from previous shocks and research emerging from the COVID-19 pandemic.[55 56] We found the need for greater flexibility, with engagement with a more diverse range of suppliers and greater decentralised control over supply chain across both settings. This is in keeping with a recent systematic review of supply chain resilience literature, which identified the importance of diversity and the social aspects of supply chains during a pandemic response.[55] Supplying commodities without investing in health systems strengthening will not produce a robust supply chain, limiting ability to respond quickly and effectively to future demands.[55]

We found a strong focus on the need for support for the health workforce, particularly in UK (principle 5). This was not as widely discussed in Liberia (though this may be a limitation relating to differing levels of participants

between countries). However, a previous study in Sierra Leone and Liberia, highlighted that many providers may carry unresolved trauma from earlier shocks (including the Ebola epidemic), which may have implications for them during the COVID-19 response.[57 58] Research among health workers treating patients with COVID-19 in China, revealed health workers had a higher prevalence of insomnia, anxiety, depression, somatisation and obsessive–compulsive symptoms compared with nonmedical health workers, indicating the need for support and recovery programmes for these staff.[59] Stressors identified among workers in China, include many of those described by participants in both settings within our study, particularly within Merseyside, including difficulties feeling safe at work, lack of IPC measures and COVID-19 knowledge, long-term workload, high risk of exposure to COVID-19, shortage of PPE and lack of rest, among others.[59]

Our findings regarding resource flow to front-line providers (principle 6), are in keeping with previous study which identified funding as a core dimension within a health systems' ability to adapt and respond to shocks.[60] A recent systematic review found aggregate public spending for health is associated with improved life expectancy, reduced child and infant mortality and more equitable health outcomes.[56]

### Relational and teamwork components

The relational components which exist are shaped by risk, trust, values, power, norms and culture.[42] These components play a role in determining the success (or failure) in response to a health systems shock or crisis. In contrast to the FCDO recommendation for good communication between actors (principle 4), our findings highlight challenges, particularly in the UK, where communication channels struggled to keep pace with changing guidance creating contradictory messaging and confusion among health workers. This is in keeping with previous study which found differences in lines of authority and acceptability of communication pathways can contribute to problems in communication.[34] In response, key principles were identified including participation for all, respect, information sharing, collaboration and problem-solving.[34]

The need for strong governance structures and leadership which adapts to the response (principle 10), was identified as a gap within early response in Merseyside. This was felt to have been hindered by slow and centralised guidance and decision-making with a perceived limited scope for autonomy within decision-making at lower levels. Within Liberia learning from the EVD response, and establishing an IMS (led by the Minister of Health) and SPACOC (led by the President) early in planning their pandemic response enabled timely decision-making.[27] In both settings, there was interest in greater decentralisation of decision-making to lower levels. Blanchet et al emphasised the need for legitimacy within resilience, with requirement of capacity to develop socially and contextually accepted institutions and norms.[40]

Looking more broadly, the conceptual framework highlights community engagement, with the community being active participants of any health systems response (principle 3).[39] Our findings emphasise the value of community engagement within the response within Liberia, based on lessons from the EVD pandemic and in keeping with WHO recommendation that this be a key pillar within COVID-19 country response.[8] Liberians across all sociodemographic groups responding to a recent survey said they were very well, or somewhat well informed about the COVID-19 pandemic, with only 5% feeling not very well/not at all informed.[27] This also emerged as a key finding in Singapore, with engagement through new and social media channels monitored, with clarification of misinformation by Ministry of Health.[61] In contrast to the findings from Liberia, participants from Merseyside highlighted the need for stronger communication (although there were some examples of creative ways to engage with diverse communities).

Learning from our study has emphasised the need to better prepared for, and respond to, health emergency crises through integrated services (principle 9).[44] A recent survey found most of the population felt the Liberian government was doing well in managing the pandemic.[44] This contrasted with findings from the UK where there was felt to have been a lack of adequate advance planning and preparation. Two previous literature reviews highlighted that 'preparedness depends on health systems ability to learn from prior pandemics', with responses often reactive rather than proactive.[56 62]

The people-centred approach stresses the need for awareness and recognition of the interdependencies of the health system with the community and other social systems, including education, social protection and food security and their relationship with social determinants of health (principle 8).[63] Our findings emphasise the need for strong partnerships with other sectors across settings, in keeping with an identified success in Singapore's response,[61] and is a key aspect of Blanchet et al's resilience framework, ensuring the capacity to engage with, and handle, multiple actors and dynamics.[40]

Our findings, particularly from Merseyside emphasise the vast quantities of data being generated through the COVID-19 response, but there are gaps in how these data are analysed and utilised within the health system. The importance of adequate HIS is in keeping with previous studies.[40 60] A health system's ability to identify and respond to an emerging threat is needed if it is to appropriately meet emerging needs during a rapidly evolving health crisis or shock (principle 7).[40 41] A robust health management information system is crucial to a health systems capacity to respond to shock.[60] Health systems need to have the ability to combine and integrate different forms of knowledge and to anticipate and cope with uncertainties and unplanned events.[40]

COVID-19 has reflected and exacerbated existing social inequalities and emphasised the importance of global collective action, rather than an individual response for genuine resilience.[8] Vaccine inequity and a lack of global solidarity on the part of some richer countries, are dominating the current phase of the pandemic. Our findings seek to highlight opportunity for shared learning across settings in the Global South and North, emphasising the need for a global response to this and future shocks.

## Strengths and limitations

The strengths of this study include the quality of data analysis, which involved a wide range of researchers across both settings, and the breadth of perspectives captured from front-line staff and key decision-makers early in the course of the pandemic. Our study had a number of limitations. Within Merseyside, study participants were selected from across a range of health system levels including primary care, hospital front-line workers and decision-makers, as well as regional decision-makers. By contrast, in Liberia participants included national and county level decision-makers, technicians and supervisors of front-line staff, with no direct front-line workers included. This may result in some of the differences in findings, related to these differing perspectives. Perhaps the greatest limitation of this study is that it was carried out at a single point in time. In Merseyside, we collected data towards the end of the first wave, at a time when there were few inpatients and people were reflecting on the first wave. Meanwhile in Liberia, it was carried out before there had been a large increase in cases. Since the study was carried out there have been subsequent even greater waves of cases within Merseyside, UK and Liberia has experienced a large surge in cases of the delta variant (59% of cases recorded in Liberia up until 17 July 2021, occurred during a 6-week period from 1 June 1 2021 to 17 July 2021).[64] By the weeks beginning 24 July 2021 to 7 August 2021, number of confirmed cases had declined between 0 and 43. Response measures have evolved in both settings, and limitations identified through the study may have been addressed in subsequent stages of the pandemic.

## Conclusion

We found the ability of health systems to be able to absorb, adapt and transform in response to the COVID-19 pandemic, in two very different settings, closely relates to the eight FCDO principles of resilience.[16 40] We expanded these principles to include strong structures and mechanisms for advance preparation, and adaptable governance and leadership structures to facilitate timely decision-making and response coordination. At the heart of our findings lies the centrality of the people-centred health system, where the person, is placed within their family, community and the health system.[14] When all aspects work together the outcome is the extent of resilience demonstrated within a health system in response to shock.[40] This includes both the provision of specific services in response to the shock experienced, as well as continued provision of, and demand for, 'routine care'.

Our study highlights the need to maintain a people-centred approach for a resilient health system response.

**Author affiliations**
[1]Department of International Public Health, Liverpool School of Tropical Medicine, Liverpool, UK
[2]Actions Transforming Lives, Monrovia, Liberia
[3]Centre for Social Ethics & Policy, School of Law, The University of Manchester, Manchester, UK
[4]Neglected Tropical Disease Programme, Ministry of Health, Monrovia, Liberia
[5]Centre for Capacity Research, Liverpool School of Tropical Medicine, Liverpool, UK
[6]Institute of Population Health, University of Liverpool, Liverpool, UK
[7]Pacific Institute for Research and Evaluation, University of Liberia, Monrovia, Liberia
[8]Liverpool School of Tropical Medicine, Liverpool, UK
[9]Department of Planning, Policy and M&E, Ministry of Health, Monrovia, Liberia
[10]Tropical Infectious Diseases Institute, Liverpool University Hospitals Foundation Trust, Liverpool, UK

**Twitter** Rosalind McCollum @RoziMcC, Zeela Zaizay @FZZaizay, Laura Dean @Laura_Deano, Imelda Bates @ImeldaBates_, Rebecca Harris @RebeccaVHarris, Shahreen Chowdhury @shahreen_c, Hannah Berrian @hannah_berrian, John Solunta Smith @soluntasmith, Wede Seekey Tate @TateWede, Taghreed El Hajj @TAG_H, Kim Ozano @Kim_Ozano, Georgina Zawolo @gvkz1, Yan Ding @YanDing9, Russell Dacombe @RussellDacombe, Miriam Taegtmeyer @MiriamTaegtmeye and Sally Theobald @sallytheobald

**Acknowledgements** We would like to thank all the participants who made time to share their experience and reflections to make this research possible. We recognise and thank Abiola Aiyenigba, who sadly passed away during the study, for her inputs. We thank Tim Martineau, and Joanna Raven (LSTM) for early inputs into study design, Susie Crossman for managing the study budget and Sue Grice for proofreading the paper.

**Contributors** Study guarantors RM and ZZ; RM prepared the first draft of the paper with inputs from all; Study design, conceptualisation, ethics (ST, LD, MT, LF, IB, ZZ, RM, VW, HP, RAdC, RH, KK); conducted interviews in UK—RM, VW, MT, KO, HP, SC, ST, TEH, RH, RD, YD, OH; conducted interviews in Liberia—ZZ, WST, HB, JK, JSS, CP, GZ, RM. All interviewers participated in the cross-country analysis which was led by YA in the UK with inputs from those who conducted UK interviews and LD, RM, ZZ, HB, WST, JK, JSS, GZ, CP in Liberia. All authors were involved in critical review of the approach, inputted into and approved the final draft of the manuscript.

**Funding** This research was funded by the NIHR Health Protection Research Unit in Emerging and Zoonotic Infections, and the Centre of Excellence in Infectious Diseases Research, and the Alder Hey Charity. We also acknowledge support of Liverpool Health Partners and the Liverpool-Malawi-Covid-19 Consortium. Additional funding to support this work came from the NIHR REDRESS Programme (NIHR2001129) and FCDO COUNTDOWN (PO6407) programme.

**Competing interests** None declared.

**Patient and public involvement** Patients and/or the public were not involved in the design, or conduct, or reporting, or dissemination plans of this research.

**Patient consent for publication** Not applicable.

**Ethics approval** Ethical approval was received from the Liverpool School of Tropical Medicine Research Ethics Committee (Protocol ID 20-045); the University of Liverpool Ethics Committee (Reference 7811) and the University of Liberia-Pacific Institute for Research and Evaluation Institutional Review Board; National Health Service Health Research Authority and Health and Care Research, Research Ethics Committee (Reference 20/HRA/2597); Integrated Research Application System (Project ID 284143).

**Provenance and peer review** Not commissioned; externally peer reviewed.

**Data availability statement** Data are available on reasonable request.

**ORCID iDs**
Rosalind McCollum http://orcid.org/0000-0003-4982-1734
Imelda Bates http://orcid.org/0000-0002-0862-8199
John Solunta Smith http://orcid.org/0000-0003-3235-0104
Yan Ding http://orcid.org/0000-0002-8439-9682

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
