## [Reviewer comments · BMJ Open]

ARTICLE DETAILS

TITLE (PROVISIONAL)	Qualitative study exploring lessons from Liberia and the UK for building a people-centred resilient health systems response to COVID-19
AUTHORS	McCollum, Rosalind; Zaizay, Zeela; Dean, Laura; Watson, Victoria; Frith, Lucy; Alhassan, Yussif; Kollie, Karsor; Piotrowski, Helen; Bates, Imelda; Anderson De Cuevas, Rachel; Harris, Rebecca; Chowdhury, Shahreen; Berrian, Hannah; Smith, John; Tate, Wede; El Hajj, Taghreed; Ozano, Kim; Hastie, Olivia; Parker, Colleen; Kollie, Jerry; Zawolo, Georgina; Ding, Yan; Dacombe, Russell; Taegtmeyer, Miriam; Theobald, Sally

VERSION 1 – REVIEW

REVIEWER	Bishai, David Johns Hopkins University, Population, Family, and Reproductive Health
REVIEW RETURNED	26-Jan-2022

GENERAL COMMENTS	General Comments I love the FCDO framework as a normative aspiration for all health systems. If it were up to me every health worker on earth would start their day by reciting all 10 principles. Furthermore, like many readers, I share an interest in what policymakers may or may have learned about system reform during the stress test of COVID-19. I am not convinced that what people have already learned is currently aligned with FCDO framework. What people actually say may not be what I wish they would say. The confusion in this paper is that in places it is committed to positivist epistemology. The aim stated on page 10 line 14 is a neutral positivist goal of describing adaptations and decision-making. Indeed, the team has created some coded transcripts based on 66 interviews from two countries and they will describe them in order to add to the sum of the world's knowledge. Sticking to that would have been grand. Except, I am actually not sure the paper is committed to positivism. Abstract says the intent is to explore the applicability of FCDO principles. In that case the focus is on the principles and the goal is the normative goal of looking at the transcripts for places to apply them. Sticking to that task, would make the paper's contribution definitely subjective, but a contribution nonetheless. The authors seem like reasonable people and if it is their opinion that FCDO principles could be applied in managing COVID-19 then readers would probably benefit from this subjective contribution. But paper has to be clear about its subjectivism. Unfortunately, the normative and descriptive tasks get muddled and that gets us into the trouble that David Hume warned us about. The paper transgresses the is/ought boundary.
--

	In the Discussion section, the leading conclusion (page 21, line 52) is that people-centred health systems are resilient. While it is my opinion that the claim is true, the evidence collected in the paper cannot prove that claim because we have no evidence that Merseyside and Liberia were exemplars of resilience. It is one thing to make normative claims as a matter of opinion, it is another to suggest that the claims have been proven by science. So the main weakness of the current version of paper is the muddle between having a neutral framework that helps readers understand the transcripts and a focus on FCDO principles whose applicability can be assessed from the interviews. If the revision pivots exclusively to the latter mission, then it needs to make it clear that we are not simply out to neutrally describe 66 interviews. The paper would also need to offer a rationale for why the FCDO principles are ipso facto so good that we will want to look for ways to apply them. I do think such a case can be made. Then take it further and be much more directive about application. FCDO principles are sterile unless someone asks and answers the question “Who is responsible for implementing the principles?” A revision would look to the transcripts for insight into ways the Liberian and UK health systems could facilitate execution of FCDO principles by building them into job descriptions, contracts, and into the organizational culture. At the moment, the analysis seems satisfied to map interview quotes to principles. For example. Page 14 line 31 has a quote that mentions the supply chain and this is taken as illustrative of principle 1. Readers desperately want the interview to shed light on reforms to incentives and system structure that will improve the supply chain, build trust, foster good communication. Who will do these things post-reform and why will they do them? A related weakness is the lack of clarity about the role of the FCDO in funding the project. Revision must tell readers if they had a direct role in the design and analysis of the paper. The consensus to adopt the FCDO principles after a series of investigator workshops does seem convenient. There is also irony in that the paper earnestly espouses respect for all, and yet the funders’ principles command the most respect. Specific Comments Abstract line 11. Paper’s stated intent “...explore the applicability of FCDO principles...” really means that the paper will sell a foregone conclusion that FCDO principles are absolutely wonderful. This stated intent is not consistent with the more open-ended aim stated on Page 10 Line 13: “To understand COVID-19 adaptations...” Abstract line 55. “..people centred approach, which places the person at the centre of study and analysis of the health system...” Please reconsider the word choice here. I hope the authors did not intend to imply that “study and analysis” is the predicate of the health system or of people-centredness. Yes, resilient health systems must study and analyze. But to confine the people-centeredness to that domain shows a reluctance to actually share power with the people. People become objects of study, not authors of their collective destiny. Page 6 Line 6. “It’s” is misspelled many times in the paper. Page 7 line42-43. I guess it is not obvious to the whole world that the severe lack of COVID testing in LMICs invalidates cross country comparisons of COVID-19 cases and death burden whenever comparisons include countries with severe under-reporting. It is not even OK to assume that rates of under-reporting are consistent
--	--

	across LMIC countries. Do not try to say “We acknowledge that data are under-reported, but Liberia is still showing some great numbers.” No conclusion about Liberia’s relative success with COVID can be drawn for now. Maybe in a year or so, the age pyramids will show the magnitude of all cause mortality by age and we can draw some conclusions based on household surveys and census data. UK does have adequate testing and its cumulative COVID-19 death count per capita from Jan 1, 2020 to Jan 1, 2022 puts it in the worst 5 countries with GDP above \$20,000. The countries chosen for this study cannot be pitched as COVID control exemplars. The UK has one of the worst COVID-19 records and data are unavailable to assess Liberia’s COVID-19 success. I would recommend that the paper pitch the country selection on the grounds that there was reason to suspect that these countries could be informative about FCDO principles because of practices they had adopted. Page 11 line 60/ Page 12 line 4. Authors should re-contemplate this part of the paper. Please consider whether it is really mathematically possible for a national level government to directly carry out “people-centred” activities without the intermediation of either local level sub national governments or private contractors at local level. There are 5 million people in Liberia. How can a few thousand people in Monrovia’s health ministry offer them any sort of inclusion in health system governance? To say that the “demand for research was focused at the national level” is probably only part of the answer. There are 15 counties in Liberia where county health officials probably would have demanded research on FCDO principles had they been asked. In a revision please alert readers to the ultimate need for sub-national government involvement in FCDO principles. Failing to do this could make national governments declare that they embrace people-centredness without including a strategy for the counties and boroughs. Page 12 line 52-56 A revision should offer much more detail about the conduct of the data analysis workshops. Given the FCDO funding source, it seems somewhat convenient that the framework that emerged from the workshop just happened to stress the FCDO framework that was “jointly developed”. (Page 10, line 29.) Page 13 line 3 Says “Most of the emerging themes aligned closely with the FCDO principles and were mapped accordingly” This is a result, not a method. Page 21 line 52 “Our findings indicate that a resilient health system...” This begs the question, “How do we know these cases were resilient health systems?” What is the evidence that the interview statements were coming from a place of resilience. Neither UK nor Liberia has credentials as being COVID-19 resilient.. Page 23 “We found...” Page 25 “Learning from our study...” These are examples where the paper purports to be undertaking a positivist contribution to knowledge. But positivism has been compromised because the paper was pre-ordained to find that the FCDO principles were praiseworthy and applicable. Page 26 Line 41-59 Reads like a Hyde Park soapbox. Please revise to tether claims to the evidence that has been collected. Page 28 line 38. Revise to include statements about the role of FCDO funders in the paper. Did they review it prior to submission? Did they have editorial rights to content? Were they co-authors?
--	---

REVIEWER	Bishop, Simon Nottingham University
REVIEW RETURNED	02-Mar-2022

GENERAL COMMENTS	Many thanks for the chance to read what in my view is a well-developed paper. Although comparing the pandemic response of a city region with that of a country is somewhat unusual, I think the approach of comparing experiences in what are often considered ostensibly different contexts is very refreshing, is well explained and something I feel is extremely appropriate in light of the global Covid pandemic. It is clear that health care systems in Global North have to identify ways to increasingly learn from the resilience of systems in the Global South and their experience of dealing with previous shocks. Having said this, the paper avoids simplistic suggestions of transferring 'best practice'. The paper is concisely written and includes a relevant framework, which is used to structure findings and draw out relevant insights from what is an inevitably complex situation. The weaknesses and limitations of the research are clearly identified. The comments below are minor points. Pg 8 'both countries have a commitment to the development of people-centred health systems' – does this refer to explicit public policy? [as many would suggest the UK NHS is not people centred in practice] Population of Liverpool is included; Liberia isn't I would suggest given the relatively unusual comparison, an explanation for the choice could be briefly provided (e.g. research team pre-existing networks facilitating timely access?) I think the table (pgs 14, 15 and 16) presenting key quotes would benefit from a headline about the broader finding of comparison/similarity/difference which the quote is illustrating. E.g. 'Shared need for decentralised procurement' (Principle 1). Or e.g. 'Building community trust' vs 'Widening mistrust' (principle 3) The point of comparison (top of pg5) comparing the communication responsiveness, should perhaps provide the caveat that comparing a city and a country, there is clearly differences in the structure of decision-making authority of those included in the study (although this is covered in the explicit limitation section)
---

VERSION 1 – AUTHOR RESPONSE

Reviewer: 1

Dr. David Bishai, Johns Hopkins University

Comments to the Author:

General Comments

I love the FCDO framework as a normative aspiration for all health systems. If it were up to me every health worker on earth would start their day by reciting all 10 principles. Furthermore, like many readers, I share an interest in what policymakers may or may have learned about system reform during the stress test of COVID-19. I am not convinced that what people have already learned is currently aligned with FCDO framework. What people actually say may not be what I wish they would say.

The confusion in this paper is that in places it is committed to positivist epistemology. The aim stated on page 10 line 14 is a neutral positivist goal of describing adaptations and decision-making. Indeed, the team has created some coded transcripts based on 66 interviews from two countries and they will describe them in order to add to the sum of the world's knowledge. Sticking to that would have been grand.

Except, I am actually not sure the paper is committed to positivism. Abstract says the intent is to explore the applicability of FCDO principles. In that case the focus is on the principles and the goal is the normative goal of looking at the transcripts for places to apply them. Sticking to that task, would

make the paper's contribution definitely subjective, but a contribution nonetheless. The authors seem like reasonable people and if it is their opinion that FCDO principles could be applied in managing COVID-19 then readers would probably benefit from this subjective contribution. But paper has to be clear about its subjectivism.

Unfortunately, the normative and descriptive tasks get muddled and that gets us into the trouble that David Hume warned us about. The paper transgresses the is/ought boundary.

In the Discussion section, the leading conclusion (page 21, line 52) is that people-centred health systems are resilient. While it is my opinion that the claim is true, the evidence collected in the paper cannot prove that claim because we have no evidence that Merseyside and Liberia were exemplars of resilience. It is one thing to make normative claims as a matter of opinion, it is another to suggest that the claims have been proven by science.

So the main weakness of the current version of paper is the muddle between having a neutral framework that helps readers understand the transcripts and a focus on FCDO principles whose applicability can be assessed from the interviews. If the revision pivots exclusively to the latter mission, then it needs to make it clear that we are not simply out to neutrally describe 66 interviews. The paper would also need to offer a rationale for why the FCDO principles are ipso facto so good that we will want to look for ways to apply them. I do think such a case can be made.

Then take it further and be much more directive about application. FCDO principles are sterile unless someone asks and answers the question "Who is responsible for implementing the principles?" A revision would look to the transcripts for insight into ways the Liberian and UK health systems could facilitate execution of FCDO principles by building them into job descriptions, contracts, and into the organizational culture. At the moment, the analysis seems satisfied to map interview quotes to principles. For example. Page 14 line 31 has a quote that mentions the supply chain and this is taken as illustrative of principle 1. Readers desperately want the interview to shed light on reforms to incentives and system structure that will improve the supply chain, build trust, foster good communication. Who will do these things post-reform and why will they do them?

A related weakness is the lack of clarity about the role of the FCDO in funding the project. Revision must tell readers if they had a direct role in the design and analysis of the paper. The consensus to adopt the FCDO principles after a series of investigator workshops does seem convenient. There is also irony in that the paper earnestly espouses respect for all, and yet the funders' principles command the most respect.

We appreciate reviewer 1's comment relating to presenting more findings relating how to facilitate execution of the principles. As outlined in the analysis section (p.11, lines 229-254) we began with inductive analysis of emerging themes. Since the FCDO principles were applied retrospectively, due to their noted applicability during analysis, we did not include probes into their application when conducting the research and so this was not expressly explored through the study. Where best practices have been described, these have been included within the results. The authors have also sought to identify recommendations which emerge from the findings in box 2. Furthermore, a second paper is under review, which described in more depth three case studies documenting best practices within study settings.

Other aspects of comments outline above have been addressed in response to the specific comments provided, as described below.

Specific Comments

Abstract line 11. Paper's stated intent "...explore the applicability of FCDO principles..." really means that the paper will sell a foregone conclusion that FCDO principles are absolutely wonderful. This stated intent is not consistent with the more open-ended aim stated on Page 10 Line 13: "To understand COVID-19 adaptations..."

We do not seek to sell a foregone conclusion relating to the FCDO principles, as we have now clarified further within the methodology, these were identified and applied following data collection and

after the initial individual country analysis, with the principles applied in order to assist with framing cross- country analysis and comparison. Thank you for these insights and we have also revised the abstract accordingly.

Abstract line 55. “..people centred approach, which places the person at the centre of study and analysis of the health system...” Please reconsider the word choice here. I hope the authors did not intend to imply that “study and analysis” is the predicate of the health system or of people-centredness. Yes, resilient health systems must study and analyze. But to confine the people-centeredness to that domain shows a reluctance to actually share power with the people. People become objects of study, not authors of their collective destiny.

We have reviewed this statement to ensure greater clarity of meaning.

Page 6 Line 6. “It’s” is misspelled many times in the paper.

Revised. Thank you.

Page 7 line42-43. I guess it is not obvious to the whole world that the severe lack of COVID testing in LMICs invalidates cross country comparisons of COVID-19 cases and death burden whenever comparisons include countries with severe under-reporting. It is not even OK to assume that rates of under-reporting are consistent across LMIC countries. Do not try to say “We acknowledge that data are under-reported, but Liberia is still showing some great numbers.” No conclusion about Liberia’s relative success with COVID can be drawn for now. Maybe in a year or so, the age pyramids will show the magnitude of all cause mortality by age and we can draw some conclusions based on household surveys and census data.

We have reviewed the statements which describe population numbers and reported COVID-19 case numbers. We agree with the reviewer’s comment that it is not possible to compare case numbers between both settings due to differences in the roll out and uptake of COVID-19 testing between settings, and so we have added a statement to make this challenge with comparison clearer (see page 6).

UK does have adequate testing and its cumulative COVID-19 death count per capita from Jan 1, 2020 to Jan 1, 2022 puts it in the worst 5 countries with GDP above \$20,000.

The countries chosen for this study cannot be pitched as COVID control exemplars. The UK has one of the worst COVID-19 records and data are unavailable to assess Liberia’s COVID-19 success. I would recommend that the paper pitch the country selection on the grounds that there was reason to suspect that these countries could be informative about FCDO principles because of practices they had adopted.

It was not our intention to say that Liverpool or Liberia are COVID-19 exemplars in how they have responded. Rather we have sought to use a pragmatic approach to highlight lessons and opportunities for shared learning from both successes and weakness in the health systems responses to COVID-19 across two very different settings. We have rephrased our text accordingly to emphasise this more clearly on page 5. Likewise, we do not feel that both settings are the epitome of people centred health systems, and so have removed a sentence which could be read to imply this – thank you for this clarification. As per our response directly below, we have made the rationale behind the choice of these two settings more explicit.

Page 11 line 60/ Page 12 line 4. Authors should re-contemplate this part of the paper. Please consider whether it is really mathematically possible for a national level government to directly carry out “people-centred” activities without the intermediation of either local level sub national governments

or private contractors at local level. There are 5 million people in Liberia. How can a few thousand people in Monrovia's health ministry offer them any sort of inclusion in health system governance? To say that the "demand for research was focused at the national level" is probably only part of the answer. There are 15 counties in Liberia where county health officials probably would have demanded research on FCDO principles had they been asked. In a revision please alert readers to the ultimate need for sub-national government involvement in FCDO principles. Failing to do this could make national governments declare that they embrace people-centredness without including a strategy for the counties and boroughs.

Liverpool (UK) and Liberia have been selected for comparison due to strong pre-existing links and networks within both settings, along with demand for research within both contexts – at the national level in Liberia and within the regional level in Liverpool. Due to these prior relationships and strong demand for research, the research team were able to rapidly carry out the study, providing more timely sharing of findings with stakeholder within both settings. We recognise that comparison of findings from health workers and regional stakeholders within Liverpool City with those from national and county level within Liberia brings a series of limitations, which we have sought to acknowledge. We agree with the need for sub-national involvement within Liberia. Our study did include some participants from county as well as national level, although we recognise that it would have been preferable to have included more participants from a broader range of health systems levels and that country, district and community engagement is critical to person centred approaches. While these limitations have been included within the paper, we have clarified these further and have moved these earlier in recognition of the importance of acknowledging and addressing these (see page 5 and page 10). Despite this limitation, we feel there remain important learnings across both settings, which we have sought to share more widely within this paper.

Page 12 line 52-56 A revision should offer much more detail about the conduct of the data analysis workshops. Given the FCDO funding source, it seems somewhat convenient that the framework that emerged from the workshop just happened to stress the FCDO framework that was "jointly developed". (Page 10, line 29.)

Page 13 line 3 Says "Most of the emerging themes aligned closely with the FCDO principles and were mapped accordingly" This is a result, not a method.

Page 21 line 52 "Our findings indicate that a resilient health system..." This begs the question, "How do we know these cases were resilient health systems?" What is the evidence that the interview statements were coming from a place of resilience. Neither UK nor Liberia has credentials as being COVID-19 resilient..

Page 23 "We found..." Page 25 "Learning from our study..."

These are examples where the paper purports to be undertaking a positivist contribution to knowledge. But positivism has been compromised because the paper was pre-ordained to find that the FCDO principles were praiseworthy and applicable.

Page 28 line 38. Revise to include statements about the role of FCDO funders in the paper. Did they review it prior to submission? Did they have editorial rights to content? Were they co-authors?

The study has sought to use a pragmatic approach to research, working through existing networks to carry out timely research to support the ongoing COVID-19 response in both settings. We have also blended both inductive and deductive aspects to our research, in keeping with applied qualitative analysis and including in health and health systems research [1,2]. The initial analysis carried out in both settings involved familiarisation and immersion in the data to identify emerging themes. This was carried out through separate data analysis workshops in each setting. Following review of the initial themes which emerged there was found to be strong similarity with the eight FCDO principles. These principles were then applied to assist with comparison between settings. We have provided further clarification relating to the data analysis workshops in each setting and jointly as on page 11.

We did not however simply accept these eight principles, we reviewed them and found that they did not fully cover all the aspects of resilience which emerged from our data. As a result we identified two further principles, which we then duly applied to adequately compare findings between both settings. The application of the expanded FCDO principles helped to showcase how Liberia's experience with responding to prior shocks and their learned need for early preparedness provided an important element working towards resilience. We would also like to clarify that this study is not funded by FCDO, nor were FCDO involved in any way as researchers or co-authors within the research team. This has been clarified on page 12.

Page 26 Line 41-59 Reads like a Hyde Park soapbox. Please revise to tether claims to the evidence that has been collected.

We have reviewed the text and made some edits to ensure that it more closely relates to the data. We have not cut this completely. At the time of our research the COVID-19 vaccine had not yet been developed and we feel that it is appropriate to situate our learning within current and ongoing global inequities.

Reviewer: 2

Dr. Simon Bishop, Nottingham University

Comments to the Author:

Many thanks for the chance to read what in my view is a well-developed paper. Although comparing the pandemic response of a city region with that of a country is somewhat unusual, I think the approach of comparing experiences in what are often considered ostensibly different contexts is very refreshing, is well explained and something I feel is extremely appropriate in light of the global Covid pandemic. It is clear that health care systems in Global North have to identify ways to increasingly learn from the resilience of systems in the Global South and their experience of dealing with previous shocks. Having said this, the paper avoids simplistic suggestions of transferring 'best practice'. The paper is concisely written and includes a relevant framework, which is used to structure findings and draw out relevant insights from what is an inevitably complex situation. The weaknesses and limitations of the research are clearly identified. The comments below are minor points.

Pg 8 'both countries have a commitment to the development of people-centred health systems' – does this refer to explicit public policy? [as many would suggest the UK NHS is not people centred in practice]

We have removed this statement.

Population of Liverpool is included; Liberia isn't

Population of Liberia is now presented on page 5.

I would suggest given the relatively unusual comparison, an explanation for the choice could be briefly provided (e.g. research team pre-existing networks facilitating timely access?)

The point of comparison (top of pg5) comparing the communication responsiveness, should perhaps provide the caveat that comparing a city and a country, there is clearly differences in the structure of decision-making authority of those included in the study (although this is covered in the explicit limitation section)

Many thanks, further clarity about this has been added. Liverpool (UK) and Liberia have been selected for comparison due to strong pre-existing links and networks within both settings, along with demand for research within both contexts. Due to these prior relationships and strong demand for research, the research team were able to rapidly carry out the study, providing timely sharing of

findings with stakeholder within both settings. We recognise that comparison of findings from health workers and regional stakeholders within Liverpool City with those from national and county level within Liberia brings a series of limitations, which we have sought to acknowledge. We agree with the need for sub-national involvement within Liberia. Our study did include some participants from county as well as national level, although we recognise that it would have been preferable to have included more participants from a broader range of health systems levels. While these limitations have been included within the paper, we have clarified these further and have moved these earlier in recognition of the importance of acknowledging and addressing these (see page 5 and page 10). Despite this limitation, we feel there remain important learnings across both settings, which we have sought to share more widely within this paper.

I think the table (pgs 14, 15 and 16) presenting key quotes would benefit from a headline about the broader finding of comparison/similarity/difference which the quote is illustrating. E.g. 'Shared need for decentralised procurement' (Principle 1). Or e.g. 'Building community trust' vs 'Widening mistrust' (principle 3)

We thank reviewer 2 for their recommendation to strengthen table 2 and we have sought to review this in response, by adding an additional column which provides headlines about the broader finding of comparison/similarity/difference which the quote is illustrating. We thank the reviewer for this valuable recommendation.

References

1. Ritchie J, Lewis J, Nicholls CM, et al. The Foundation of Qualitative Research. Qual Res Pract A Guid Soc Sci Students Res. 2013;0–25.
2. Gale NK, Heath G, Cameron E, et al. Using the framework method for the analysis of qualitative data in multi-disciplinary health research. BMC Med Res Methodol [Internet]. 2013 Jan [cited 2014 May 1];13(1):117. Available from: <http://www.pubmedcentral.nih.gov/articlerender.fcgi?artid=3848812&tool=pmcentrez&rendertype=abstract>

VERSION 2 – REVIEW

REVIEWER	Bishai, David Johns Hopkins University, Population, Family, and Reproductive Health
REVIEW RETURNED	01-May-2022
GENERAL COMMENTS	The revised paper has improved the presentation of the study findings and is set to make a great contribution.
REVIEWER	Bishop, Simon Nottingham University
REVIEW RETURNED	30-May-2022
GENERAL COMMENTS	Many thanks for the opportunity to review this revised paper. The authors have made changes in response to the reviewers comments, which help to clarify the assumptions underpinning the study and also bring out findings. Having said this, the authors may wish to consider again the fit between the aims identified 170-174, the expanded description of data analysis (which identifies that the FCDO principles were identified as potentially useful during analysis, rather than the foundations for the research) and the aims conveyed in the 'introduction' section of the abstract to ensure consistency. Further, as additional principles for resilience are added to the

	FCDO principles, perhaps it would make sense to refer to these as 'expanded principles for resilience' (e.g. lines 274, 420) as current wording may suggest the additional principles have been taken up. Otherwise in my view the findings make a contribution to knowledge and recommendations for practice.
--	--

VERSION 2 – AUTHOR RESPONSE

Comments to the Author:

Many thanks for the opportunity to review this revised paper. The authors have made changes in response to the reviewers comments, which help to clarify the assumptions underpinning the study and also bring out findings. Having said this, the authors may wish to consider again the fit between the aims identified 170-174, the expanded description of data analysis (which identifies that the FCDO principles were identified as potentially useful during analysis, rather than the foundations for the research) and the aims conveyed in the 'introduction' section of the abstract to ensure consistency.

Many thanks, we have revised the abstract for improved consistency to include the study aim within the introduction, and have clarified that the FCDO principles were found to provide a valuable framework during the process of the analysis of findings.

Further, as additional principles for resilience are added to the FCDO principles, perhaps it would make sense to refer to these as 'expanded principles for resilience' (e.g. lines 274, 420) as current wording may suggest the additional principles have been taken up. Otherwise in my view the findings make a contribution to knowledge and recommendations for practice.

Many thanks, we have reviewed and modified the text throughout the paper to refer to 'expanded principles for resilience'.